# Optimization of Imaging Reconnaissance Systems Using Super-Resolution: Efficiency Analysis in Interference Conditions

**DOI:** 10.3390/s24247977

**Published:** 2024-12-13

**Authors:** Marta Bistroń, Zbigniew Piotrowski

**Affiliations:** Institute of Communication Systems, Faculty of Electronics, Military University of Technology, 00-908 Warsaw, Poland; zbigniew.piotrowski@wat.edu.pl

**Keywords:** faster R-CNN, imaging reconnaissance, object detection, Real-ESRGAN, super-resolution

## Abstract

Image reconnaissance systems are critical in modern applications, where the ability to accurately detect and identify objects is crucial. However, distortions in real-world operational conditions, such as motion blur, noise, and compression artifacts, often degrade image quality, affecting the performance of detection systems. This study analyzed the impact of super-resolution (SR) technology, in particular, the Real-ESRGAN model, on the performance of a detection model under disturbed conditions. The methodology involved training and evaluating the Faster R-CNN detection model with original and modified data sets. The results showed that SR significantly improved detection precision and mAP in most interference scenarios. These findings underscore SR’s potential to improve imaging systems while identifying key areas for future development and further research.

## 1. Introduction

Image reconnaissance involves gathering information using various imaging technologies and sensors. Currently, satellite systems, aerial reconnaissance platforms, and unmanned aerial vehicles, supported by diverse optical tools, radars, and lasers, are most commonly used for this purpose [1]. Imaging has become a critical component in modern applications due to technological advances that have greatly improved techniques for capturing and transmitting images. These systems are widely used in fields such as environmental monitoring, disaster response, and infrastructure analysis, where precise and timely information is essential [2,3].

The speed and efficiency of image reconnaissance systems are crucial for enabling real-time decision-making and accurate analysis. However, these systems face numerous challenges that can significantly impact their effectiveness. Maintaining high detection and identification efficiency in disrupted environments ensures reliable performance and enhances situational awareness [4]. Factors causing disturbances in image reconnaissance systems include:Atmospheric Factors—Reduced visibility due to rain, fog, or sandstorms can hinder object detection, while artifacts from atmospheric disturbances may lead to incorrect classifications or object omissions [5,6];Deliberate Interference—Intentional disruptions can be introduced through techniques such as masking or signal interference. For example, the use of camouflage [7] can make objects less visible or entirely undetectable to imaging systems. Signal disruptions, particularly in drone-based systems, can destabilize image feeds, cause signal loss, or result in other distortions that hinder the collection of accurate data [8];Lossy Compression—Bandwidth limitations in telecommunication channels [9] necessitate lossy compression in reconnaissance systems, which, depending on the compression level, can introduce artifacts leading to detail loss. This significantly reduces detection efficiency, especially for smaller or less distinct objects [10,11];Object Dynamics—When capturing objects from long distances, e.g., using drones or satellite systems, the movement of the observed object or camera can cause image blur. This reduces contour sharpness, complicating object detection and classification [12,13].

The above disruptive factors can have significant consequences, impacting the reliability and effectiveness of imaging reconnaissance systems [14]. These include misclassification of objects, low detection efficiency, and the occurrence of numerous false alarms [15], which translates into the following:Reduced efficiency due to increased response times and delays in critical decision-making [16];Greater strain on resources, as inappropriate actions triggered by false alarms divert attention from actual priorities. This can lead to reduced confidence in the technology and a tendency to disregard alerts;Loss of informational advantage, providing opportunities for unforeseen events or delays in addressing critical situations, ultimately compromising the effectiveness of operations [17].

The malfunctioning of the reconnaissance system can even lead to tragic consequences, such as the downing of an Iranian passenger plane in 1988, which was misidentified as an F-14 fighter [18], which translates into a decrease in the sense of security and trust in modern technologies [19].

For the above reasons, reconnaissance systems, even state-of-the-art ones based on advanced artificial intelligence algorithms, can lead to misidentifications and the resulting serious operational consequences. The solution to optimize the performance of such systems and increase their efficiency, especially in disturbed conditions, is to use super-resolution (SR) technology, whose main task is to increase the resolution without loss [20]. However, modern SR algorithms not only increase the resolution of analyzed images but also, through the use of GAN architecture, enable simultaneous quality improvement.

The following work analyzes the effectiveness of the aircraft detection algorithm under simulated interference conditions, which were obtained by introducing appropriate artifacts resulting from the most common interferences occurring in real systems. The impact of the application of the Real-ESRGAN algorithm on the efficiency and correctness of the detection model, both in the evaluation and training process, was analyzed.

The main contributions of the authors are as follows:Reviewing and taxonomy of super-resolution models based on neural networks;Application of the super-resolution model in the detection of objects and analysis of its impact;Conducting a series of experiments with different types of interference under different scenarios;Evaluating the impact of SR technology on the selected detection model, identifying its limitations in a given application, and defining directions for further research.

The remainder of this paper is organized as follows: Section 2 describes the super-resolution technology, with particular emphasis on the Real-ESRGAN model and examples of use in reconnaissance systems; Section 3 presents the research methodology: the datasets used, the neural network models, the experiments conducted and a description of the test conditions; Section 4 presents the results of individual experiments with visualizations; Section 5 provides a discussion; Section 6 summarizes the previous considerations and presents the planned directions for further research.

## 2. Super-Resolution in Imaging Systems

Super-resolution in images refers to the lossless increase in image resolution from low (LR) to high (HR). The idea behind SR algorithms is to reconstruct images to high resolution based on pixels and other information from the original image. The technology is used in many areas:Medical—for medical data, capturing high-resolution images can be difficult to achieve [21]. Super-resolution can be used as part of medical image preprocessing to increase the efficiency of diagnostic algorithms [22,23,24,25];Surveillance and security systems—the use of super-resolution in surveillance produces clearer camera images, allowing for greater accuracy in identifying people or vehicles in city surveillance systems, during event analysis, or in IoT systems [26,27,28,29,30];Film and television industry—super-resolution allows for increasing the resolution of material recorded at a lower resolution and adapting it to current HD standards; it also facilitates the implementation of image and video watermarking algorithms [31,32,33] and augmented reality (AR) [34].

Modern super-resolution algorithms make it possible not only to increase the resolution of images but also to improve their visual quality by, among other things, reducing noise, sharpening, and reconstructing missing elements [35], which is particularly important in the context of optimizing military imaging reconnaissance systems.

### 2.1. Overview of Super-Resolution Algorithms

The first super-resolution methods were based on various interpolation techniques (Nearest Neighbor Interpolation [36], Bilinear Interpolation [37], Bicubic Interpolation [38]), frequency domain reconstruction [39], statistical methods [40,41], and multi-frame reconstruction [42] with the availability of multiple images of the same object. The methods were characterized by rather poor quality, especially at high magnifications, and required large computational resources.

With the development of neural networks, algorithms began to emerge to produce high-resolution images with realistic details of comparable quality to the original images. The first solution was the SR-CNN model [20] based on convolutional layers, in which the first layer was responsible for feature extraction, the second for nonlinear mapping, and the third for reconstruction to higher resolution. An improvement in this architecture was the Very Deep Super Resolution (VDSR) model, which reduced the size of the convolution filters and used gradient clipping to optimize the learning process [43].

To reduce the required computational power, the FSR-CNN solution was proposed, in which the preprocessing present in SR-CNN was abandoned; feature extraction was performed in low-resolution space, and reconstruction to high resolution was performed using deconvolution filters [44].

Further solutions in this area were based on the use of different types of neural networks. Based on residual blocks, EDSR (enhanced deep super-resolution) and MDSR (multi-scale deep super-resolution) models were developed [45], as well as CARN (Cascading residual network), which used a cascading mechanism to enable the network to process more information [46]. Using recursive networks, models have been developed for DRCN (Deep Recursive Convolutional Network), which used one and the same layer repeatedly instead of multiple convolutional layers to process features more deeply without adding more parameters [47], and its improvement, DRRN (Deep Recursive Residual Network), where residual blocks were used instead of convolutional layers [48].

To increase the scaling factor to eight times the resolution, progressive solutions have been used, which increased the resolution of images gradually (step-by-step). One such approach is LapSRN (Laplacian Pyramid Super-Resolution Network) [49].

The breakthrough was the application of the attentional mechanism. In the SelNet (Selective Network) architecture, an additional selective unit was added to the convolutional blocks, which decided which processed features were most relevant and would allow for efficient high-resolution image modeling [50]. The RCAN (Residual Channel Attention Network) architecture, on the other hand, combines residual blocks with a channel-level attentional mechanism that allows us to dynamically assign weights to different channels of image features, thus increasing the quality of reconstruction [51].

The solutions described focus exclusively on optimizing pixel-level differences in LR and HR images. However, from the perspective of the human visual system, it is more important to optimize perceptual quality. This is how generative models work, which are very often used when transforming one image into another but have been successfully applied to other types of signals as well [52,53,54]. The first approach using a generative model to increase resolution was the SRGAN (Super-resolution Generative Adversarial Network) architecture, which, by using perceptual loss functions, made it possible to generate images with a more natural appearance, clear edges, and details [55]. On this basis, further models were developed, such as ESRGAN (Enhanced Super-Resolution Generative Adversarial Network), where the perceptual loss function was improved [56], Real-ESRGAN [57] and SRResNet, where residual blocks were introduced [58], TSSRGAN (Two-Stage SRGAN) and TSSRGAN (Two-Stage SRGAN), where the resolution enhancement process was divided into two stages to achieve greater stability and accuracy [59].

A recent approach used in super-resolution models was using vision transformers [60], which were developed based on the popularity of traditional transformers. Using a large-scale attention mechanism allows us not only to increase resolution but also to remove noise and reduce blur. Recent developments of this type include Swin2SR using a Swin Transformer based on the concept of a sliding window transformer [61], its improvement Swin2-MoSE [62], and EpiMISR, which create HR images based on multiple images containing complementary information [63].

The following diagram shows the taxonomy of the super-resolution methods described above—Figure 1.

### 2.2. Real-ESRGAN Model

In the experimental part of this article, the Real-ESRGAN (Real-World Enhanced Super-Resolution Generative Adversarial Network) model was used to increase the resolution and improve the quality of data distorted with artifacts typical of real-world conditions. This choice was based on the versatile capabilities of this model, which was specifically adapted to low-quality data and operational conditions often encountered in real-world video systems.

Real-ESRGAN was developed on the basis of ESRGAN, but this model was adapted to process real data, which may be noisy, distorted by compression, blurred, and contain other artifacts due to atmospheric conditions or image capture technology. The basis of the model’s architecture—Figure 2—is RRDB (Residual-in-Residual Dense Block), which introduces hierarchical residual connections. This enables accurate modeling of image details while minimizing the effect of smoothing, which is crucial for reconstructing details in images with high levels of noise. In addition, an enhanced perceptual loss function, which evaluates the image based on features relevant to human perception, and generative loss functions were used to increase the sharpness and detail of the final image. The versatility of Real-ESRGAN, which successfully processes data from a variety of low-quality sources, makes it widely used in surveillance and security applications [64].

Real-ESRGAN was selected for further experimentation because of its ability to reconstruct detail in distorted images and its ability to reduce noise. This is crucial for image analysis in systems where the quality of input data is limited.

### 2.3. Related Works

Super-resolution is a very popular technique in imagery reconnaissance, where detailed visual data are crucial, especially with satellite or drone imagery. High resolution allows for more precise situational analysis and supports decision-making.

SR is most often used in systems where images come from satellites or drones. The data are recorded from a high altitude, often under adverse weather conditions. The paper [65] presented the MAGiGAN system, used in a satellite system for MISR (Multi-angle Imaging SpectroRadiometer) images, enabling a 3.75-fold increase in resolution while maintaining data quality. Similarly, the SRGAN-MSAM-DRC model was created [66]. It is a generative model with a multi-scale attention mechanism and residual blocks, which allows it to catch subtle details in remote-sensing images that feature complex spatial arrangements. Super-resolution technology, using various neural network models, positively influences the quality of satellite images after increasing their resolution, which is also confirmed by the following works [67,68,69,70]. SR is also used to process images from radar and thermal cameras, which are very often noisy, as shown in the works [71,72,73,74,75].

In environments where interference can significantly hinder object detection, the role of visual enhancement is particularly important [76]. For example, in [77], the use of super-resolution with adaptive regularization based on MG-MRF (Multi-Grid Markov Random Field) was proposed to aid the detection of point objects in infrared images. In another paper [78], the authors proposed an algorithm based on the YOLO-SR architecture with data preprocessing to facilitate the detection of small targets in remote sensing images, while in [79], a system with a generative model, attention mechanism, and convolutional layers for infrared images is presented.

In summary, super-resolution techniques are a crucial element in imaging systems, enhancing the quality of visual data even under challenging interference conditions [80,81].

## 3. Methodology

### 3.1. Military Aircraft Recognition Dataset

The Military Aircraft Recognition dataset imported from the Kaggle platform [82] was used for the research in the following work. This is a dataset of satellite images depicting different types of military aircraft dedicated to object detection. The set consists of 3842 images in which 22,341 instances of objects belonging to 20 different classes were detected, along with annotations. For implementation reasons, the annotation file was converted from Pascal VOC format [83] to COCO format [84] and adapted to the requirements of the detection algorithm used. In the COCO format, the annotations are stored as a JSON file, which consists of the following sections: info—contains basic information about the dataset, such as creation date, authors, etc.; images is a list of JSON objects, containing metadata about each image in the dataset; annotations is also a list of objects containing key data about each instance in the dataset; categories—a list of all object classes. The annotations define, among other things, the coordinates of the bounding boxes. Due to the requirements of the detection model, the coordinates remained in the format (xmin, ymin, xmax, ymax), which referred to the position in the image expressed in pixels.

The figure below shows an example of images from the dataset, with the bounding boxes and class identifiers highlighted—Figure 3.

Most of the images are 800 × 800 pixels. The images are characterized by different perspectives, lighting levels, surroundings, and visual quality. The marked objects vary in size; some are independent objects, some are parts of groups. Usually, several objects can be observed in the images, representing at least two classes. Such a selection of data allows us to reflect real detection challenges.

In order to ensure effective model training and evaluation, the collection was divided into three parts: training; validation; and test, with a ratio of 70:20:10.

For further experiments, 7 additional subsets were created, based on images from the test set, in which selected distortions and artifacts were introduced to realistically reflect conditions encountered in real applications, such as aerial reconnaissance. The following types of distortions were used:Gaussian blur allows for the simulation of small camera vibrations or vehicle movements during image acquisition, which typically results in isotropic blurring. This type of blur is applied using a convolution kernel with a two-dimensional Gaussian distribution, where the intensity of the blur decreases symmetrically from the center. As a result, Gaussian blur affects all directions uniformly, preserving structural details to some extent. The kernel size used was (15, 15), which reflected well the level of interference in real conditions;Gaussian noise reflects interference from sensors or distortion that occurs during data transmission, which is common in long-distance monitoring systems. The following noise distribution parameters were used: mean = 0 and std = 25 to reflect real interference conditions;Contrast reduction was used to simulate low-light conditions or the presence of fog, which affected the visibility of details;Motion blur is used to simulate a situation in which a drone or aircraft moves while taking a picture; the camera is in continuous motion, or the captured object is moving rapidly during exposure. Unlike Gaussian blur, motion blur is directional with a linear kernel applied along a specific axis (e.g., horizontal or vertical), creating a smearing effect in one direction. The kernel size used was also (15, 15), but the directional nature of motion blur caused a much more pronounced degradation of image details compared to Gaussian blur. This difference arises because motion blur results in the loss of fine structural features in the direction of motion, making the degradation more challenging for both image restoration and object detection;JPEG compression artifact is one of the most widely used image compression processes, but depending on the degree of compression, it can introduce visible artifacts such as pixel blocks or loss of detail, which makes detection especially difficult, especially for small objects;Changing the color balance can simulate problems caused by incorrect camera calibration or varying atmospheric conditions, such as different light intensities. Problems with color balance can affect object recognition by causing unnatural coloring or tonality in the image.

The effects of applying the distortions mentioned are shown below for a selected image from the dataset—Figure 4.

### 3.2. Implementation and Application of SN Models

#### 3.2.1. Detection Model

In image recognition systems, detection models play a key role in enabling precise localization and classification of objects in recorded images. In the present study, the Faster R-CNN (Region-based Convolutional Neural Network) model [85] was selected as the base algorithm, whose efficiency and effectiveness were analyzed in detail. The choice of Faster R-CNN was motivated by its well-established balance between computational complexity, which impacts processing efficiency and detection performance. This makes it particularly suitable for applications where both accuracy and processing speed are critical.

This model represents an improved version of its precursors, R-CNN [86] and Fast R-CNN [87], in terms of speed of operation and detection accuracy. The main components of its architecture are the following:Region Proposal Network (RPN)—a convolutional network responsible for generating potential regions of interest in feature maps;ROI Pooling (Region of Interest Pooling)—a mechanism that transforms potential regions of interest to a uniform size.

The final part of the architecture is a classifier based on fully connected layers and a regression layer that predicts the exact values of bounding box coordinates. A diagram of the model’s architecture is shown in Figure 5.

In this study, an implementation was used in which the underlying convolutional network was the ResNet-50 model [90], and model training was performed using the COCO (Common Object in Context) dataset [84]. Fine-tuning was performed for this model using the Military Aircraft Recognition dataset. The following hyperparameters were assumed during training:number of epochs—50;optimizer—SGD with *lr* = 0.005, momentum = 0.9, weight decay = 0.0005;batch size—train batch size = 4; validation batch size = 2.

After increasing the resolution of the training data, training of the detection model was conducted again. The training was terminated early due to the significantly increased processing time of the higher-resolution data and the lack of further improvement in the metrics, indicating that a stable level of model performance had been achieved.

#### 3.2.2. Super-Resolution Model

This study used the Real-ESRGAN model to increase the resolution of both the training data and additional subsets four times with introduced interference. To keep the results consistent, the coordinates of the bounding boxes, stored in the annotation file, were also scaled according to the increased resolution.

### 3.3. Description of Experimental Stages

The following experiments were conducted as part of this study:Training and evaluation of the detection model on the original training dataset. The detection model was trained and tested on the original data without any additional modifications or distortions. The purpose of this stage was to obtain reference values for detection metrics, which served as a baseline for assessing the impact of subsequent image modifications;Implementation of artifacts and distortions in test images and conducting detection. Artifacts were applied to the original test images, as described earlier. Object detection was then performed using the model trained in the first step. The results were compared with the reference results to evaluate how distortions affect the performance of the detection mode;Enhancing resolution and improving the quality of distorted images. A super-resolution model was applied to increase the resolution and improve the visual quality of distorted images. After image reconstruction, detection was performed again using the model from the first step. The results were compared with earlier results on images without quality improvement and with the reference values to assess whether the super-resolution technique enhanced the detection model’s performance under challenging conditions;Increasing the resolution of the original dataset and retraining the detection model. The original training images were processed using the super-resolution model to enhance their resolution. Subsequently, the Faster R-CNN model underwent fine-tuning, resulting in a new detection model. The results of the newly trained model were compared with those obtained by the baseline model from the first step to determine how higher image resolution impacted detection accuracy during the model training phase;Detection of Distorted and Restored Images Using the New Detection Model. In the final experiment, the model trained on high-resolution images (from the fourth step) was tested on distorted test images and on distorted images restored using super-resolution. The goal was to compare detection results for both sets of images—distorted and reconstructed—using the new model. This allowed for evaluating how a model adapted to high-resolution, high-quality images performs in detecting objects under distorted and restored conditions, and the impact of image reconstruction on detection effectiveness.

### 3.4. Experimental Conditions and Metrics

The experiments were conducted locally on a laptop running Windows 11, equipped with an Intel Core i9-13980HX processor operating at 2.2 GHz with 24 cores, ensuring fast data processing. GPU acceleration was utilized with CUDA 12.6, supporting an NVIDIA GeForce RTX 4090 graphics card with 16 GB of memory, enabling efficient model training and evaluation. The implementation and training of models were conducted using the PyTorch 2.5 framework, along with the following key libraries: NumPy 1.26.3; OpenCV 4.10.0.84 (preprocessing and data manipulation); pycocotools 2.0.8 (annotation management and evaluation metrics); and matplotlib 3.9.2 (data visualization).

To evaluate the performance of the super-resolution model, the PSNR (Peak Signal-to-Noise Ratio) metric was utilized, providing a quantitative assessment of image quality. PSNR, measured in decibels (dB), quantifies the difference between two images. Higher values indicate greater similarity to the original image, signifying better reconstruction quality and reduced noise or distortions. This metric is widely used for assessing the fidelity of image restoration and enhancement techniques.

To accurately evaluate the effectiveness of the object detection model, key evaluation metrics were employed, allowing for a comprehensive analysis of results:Precision—The ratio of correctly detected objects of a given class to all detected objects. A high precision value indicates a low number of false detections, which translates to minimizing false alarms;Recall—The ratio of correctly detected objects of a given class to all objects of that class that this model should have detected. A high recall value signifies that this model successfully detects most objects, which is crucial in applications such as security systems;mAP (Mean Average Precision)—This metric represents the mean of the Average Precision (AP) values calculated for each class. AP measures the area under the precision–recall curve, computed individually for each class at various levels of the IoU (Intersection over Union) parameter. IoU indicates the extent to which the predicted detection area overlaps with the actual area. This metric accounts for both precision and recall, enabling a holistic assessment of the model’s performance.

## 4. Results

The following section presents the results of individual experiments conducted in accordance with the methodology described in the previous section.

### 4.1. Training and Evaluation of the Reference Model

During the fine-tuning of this model, the efficiency of the training process was monitored based on the loss function values for the training data and the metrics values of precision, recall, and mAP for the validation data. The trends of these parameters are illustrated in Figure 6 and Figure 7. A summary of the metrics values for the test set is presented in Table 1, while a detailed interpretation of the mAP metric, including Average Precision and Average Recall for various object sizes, the number of detections per image, and IoU thresholds, is provided in Table 2. Additionally, the figures illustrate examples of the detection model’s performance on selected images from the test set—Figure 8 and Figure 9.

The results for the reference model indicate high effectiveness in object detection tasks, particularly for medium-sized objects (mAP = 0.755) and large objects (AR = 0.904). However, this model demonstrates lower precision (0.630) compared to recall (0.847), suggesting a higher number of false positives. This model provides a solid foundation for further experiments aimed at improving detection quality through super-resolution techniques.

### 4.2. Evaluation of Distorted Images

After implementing distortions into the images, this model was evaluated for each subset of data. The values of the obtained detection metrics are presented in Table 3. Additionally, for distortions with the smallest impact (Contrast reduction) and the largest impact (Motion blur) on detection performance, a detailed interpretation of the mAP metric is provided in Table 4 and Table 5. Na Figure 10 illustrates the influence of distortions on the performance of the detection model.

The research results demonstrate a significant impact of the type of distortion on the system’s performance. The smallest decrease in mAP was observed with contrast reduction (a drop to 0.583), suggesting that this model was relatively resistant to this type of distortion. In contrast, the largest drop in mAP occurred with motion blur, where the mAP value fell to 0.236. Contrast reduction mainly resulted in the loss of some details but did not prevent this model from detecting object boundaries. Motion blur, on the other hand, significantly deteriorated detection results (Figure 10), as it substantially reduced the clarity of object boundaries, leading to difficulties in detecting actual objects (low recall value) and their accurate classification (low precision value). In addition, motion blur has a much greater degrading effect than Gaussian blur despite using the same kernel size (15, 15). Directional and asymmetric motion blur produces a “smearing” effect and largely prevents the detection of object boundaries.

### 4.3. Evaluation of Images After Applying Super-Resolution

After applying the super-resolution model to the distorted images, similarity metric values were determined for the original (HR), distorted (LH), and restored (SR) images to evaluate the model’s performance for each distortion type. The detection model was then re-evaluated for each subset of the data. The metric values obtained are presented in Table 6 and Table 7. Figure 11 and Figure 12 illustrate the impact of using the super-resolution model on the detection model’s performance for selected examples.

A preliminary evaluation of the super-resolution model’s performance based on the PSNR metric showed that this model effectively reduced distortion and improved image quality, as evidenced by the high value of the metric when comparing LR and SR. The comparison of HR vs. SR also confirms that the SR images are close to the originals but differ, suggesting that the quality of the images has been improved. Based on the positive initial evaluation, further analysis was performed using the detection model.

This research demonstrated that applying the Real-ESRGAN model improved detection efficiency for all restored subsets, except for images affected by motion blur. This model could not sufficiently sharpen the images to enhance the visibility of key features and contours. For restored images distorted by contrast reduction, a recall value slightly higher than that of the original, undistorted dataset was achieved.

### 4.4. Training and Evaluation of This Model Using the Resolution-Improved Data

After applying the super-resolution model to the training dataset, fine-tuning was performed, resulting in a new detection model. Figure 13 compares the loss function trends during the training phase for both models, while Table 8 presents the detection metrics for the test set. Additionally, Table 9 provides a detailed interpretation of the mAP metric.

After fine-tuning the new model using the high-resolution data, a significant improvement in performance was observed compared to the reference model. The precision metric for the test dataset increased from 0.630 to 0.744, while the recall value decreased to 0.840. This ultimately resulted in an mAP increase to 0.701, confirming the model’s overall improved accuracy.

A detailed analysis of the metrics shows that the SR model achieves an Average Precision (AP) of 0.744 at an IoU threshold of 0.50, demonstrating its capability for effective detection under lower localization accuracy requirements. For more stringent thresholds (IoU = 0.75), the AP is 0.743, indicating the model’s stability across various scenarios.

### 4.5. Evaluation of Distorted and Restored Images

The new model was evaluated on distorted and restored images using the super-resolution algorithm. The evaluation metric values obtained are presented in Table 10.

Similar to the reference model, the introduction of distortions to the images reduced detection performance across all subsets. However, applying the super-resolution model during image preprocessing improved performance. No improvement was achieved for motion blur distortions or the combination of multiple distortions. In most cases, slightly lower recall values were observed compared to the analogous cases evaluated by the original model. However, significantly higher precision and, ultimately, higher mAP values were achieved. This suggests a potential trade-off between the model’s sensitivity and precision, which may be acceptable—or even desirable—in applications that require high precision.

## 5. Discussion

This section presents an analysis of the results obtained from the conducted experiments, considering their significance in the context of practical applications in object detection systems.

### 5.1. Limitations of This Model

The implementation of the super-resolution model (Real-ESRGAN) used in this study was unable to improve results for motion blur distortions or for a combination of distortions that included motion blur. This type of distortion significantly alters the edges and detailed features of objects, which are critical for the performance of detection models. While super-resolution enhances the visual quality of an image, it may fail to faithfully reconstruct missing details in cases of extreme blur. In such instances, the SR model may amplify distorted or erroneous patterns caused by blur, leading the detection model to interpret them as incorrect shapes and completely ignore such objects.

In the case of combined distortions, this model must address multidimensional problems that are more challenging to reconstruct than single distortions. Super-resolution models perform best with distinct distortions that can be reduced, such as noise or low resolution. For multiple distortions, SR can generate additional artifacts or erroneous details that degrade the detection model’s performance, resulting in extremely low metric values.

### 5.2. Super-Resolution Versus Data Augmentation

This research raises the question of whether a simpler solution might involve using data augmentation with selected distortions during the training of the detection model, enabling this model to learn how to localize and recognize objects affected by specific artifacts.

Data augmentation is a widely used technique in machine learning that artificially increases the size of the training dataset by introducing selected transformations and modifications to images. Despite incorporating similar modifications, augmentation and super-resolution have different goals and effects on model performance.

Augmentation introduces artifacts in a non-standard way that does not reflect real operational conditions. The generated data are often unnatural, meaning that instead of teaching this model to handle specific types of distortions, it can lead to degraded model performance due to generalization issues, ultimately resulting in lower effectiveness when analyzing real-world data.

Moreover, augmentation does not improve image quality; the modifications introduced are random and aim to increase data diversity. In extreme cases, excessive data diversity can also be undesirable, as it makes it more difficult for this model to learn patterns. This is particularly problematic in complex operational conditions, leading to reduced model efficiency.

### 5.3. Limitations of Super-Resolution Technology

While the results from laboratory studies show the potential of super-resolution models to improve image quality and increase detection performance in detection systems, there are inherent limitations and challenges to their use. One significant issue is the potential introduction of artifacts by the resolution enhancement and quality improvement process itself, which can distort the original image content and negatively impact detection accuracy. Artifacts can lead to false positives and be a source of false alarms or obscure critical details in reconstructed images, which can lead to key objects going undetected.

Additionally, tuning SR models for specific detection tasks remains a very difficult process. It requires the use of problem-specific data sets. It is both computationally intensive and requires considerable domain expertise. Solving these limitations is key to further developing the practical application of SR technology in real-world scenarios, especially in more complex applications than simply scaling images according to a specific coefficient.

## 6. Summary

The conducted research demonstrated that the application of super-resolution technology, specifically the generative Real-ESRGAN model, significantly enhanced object detection performance in image reconnaissance systems. The increase in mAP and precision in most scenarios confirmed that improving the quality and resolution of input data is critical for object detection tasks, especially under challenging conditions such as noise or lossy compression. However, the analysis also highlighted the limitations of this technology, particularly in the case of motion blur and complex combinations of distortions, where the effects of SR were less pronounced or even detrimental to detection performance. The increase in detection precision, at the cost of a minimal decrease in recall observed during the fine-tuning of the second model, is advantageous in applications where minimizing false alarms and ensuring precise detection are critical.

Proposed directions for future research are as follows:Further improvement of super-resolution models, particularly in handling extreme distortions such as motion blur, complex combinations of distortions characteristic of real-world conditions, less common types of distortions, such as occlusion or environmental factors, and distortions of varying intensity levels, to provide a more comprehensive analysis;Combining super-resolution with other image-processing techniques, such as adaptive filters, to verify whether combining SR methods with traditional image enhancement techniques provides higher detection efficiency for very degrading types of distortions;Investigating the integration of super-resolution with advanced detection algorithms, such as models incorporating attention mechanisms such as Transformer-based architectures (e.g., DETR or YOLOv7 with attention layers), and considering tuning SR models to specific research problems;Developing lightweight SR models optimized for real-time applications, especially in resource-constrained environments—optimization for computational efficiency and developing SR models specifically tailored for deployment in low-resource environments, such as edge devices, drones, or satellite systems, where power consumption and memory are limited.

This research provides a solid foundation for the continued development of super-resolution technology in the context of image reconnaissance systems while highlighting areas requiring further optimization and exploration.

## Figures and Tables

**Figure 1 sensors-24-07977-f001:**
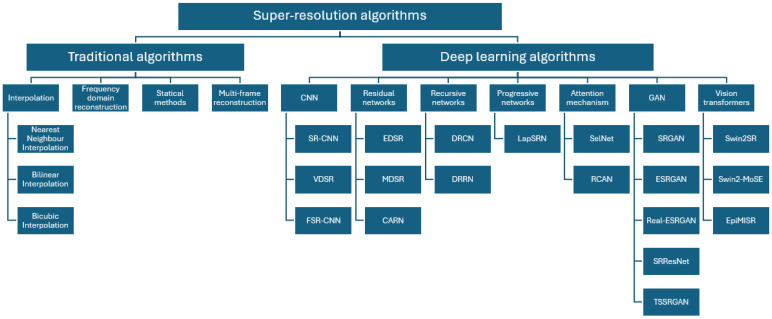
Super-resolution algorithms taxonomy.

**Figure 2 sensors-24-07977-f002:**
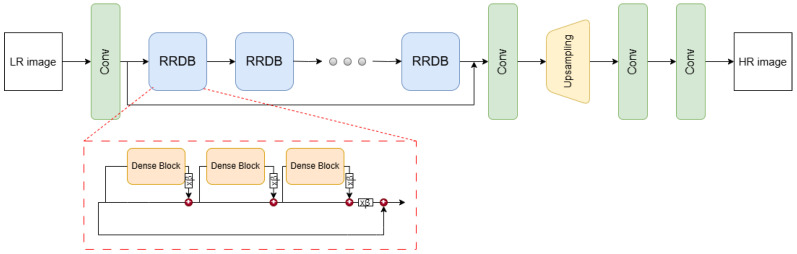
Real-ESRGAN architecture [57].

**Figure 3 sensors-24-07977-f003:**
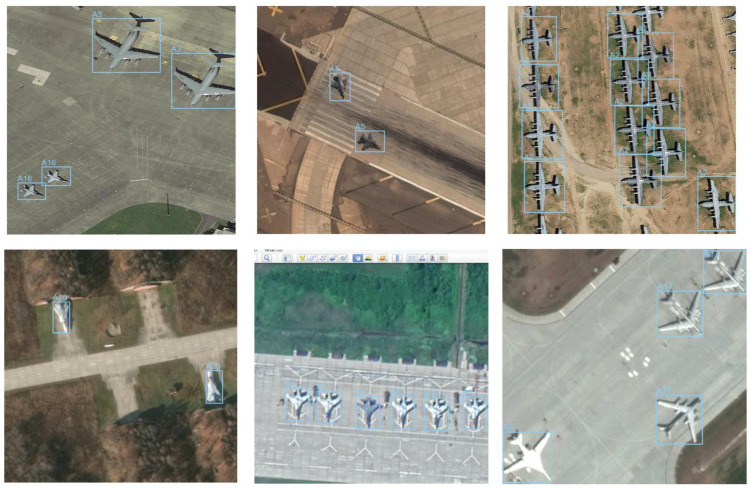
Example images from the Military Aircraft Recognition dataset.

**Figure 4 sensors-24-07977-f004:**
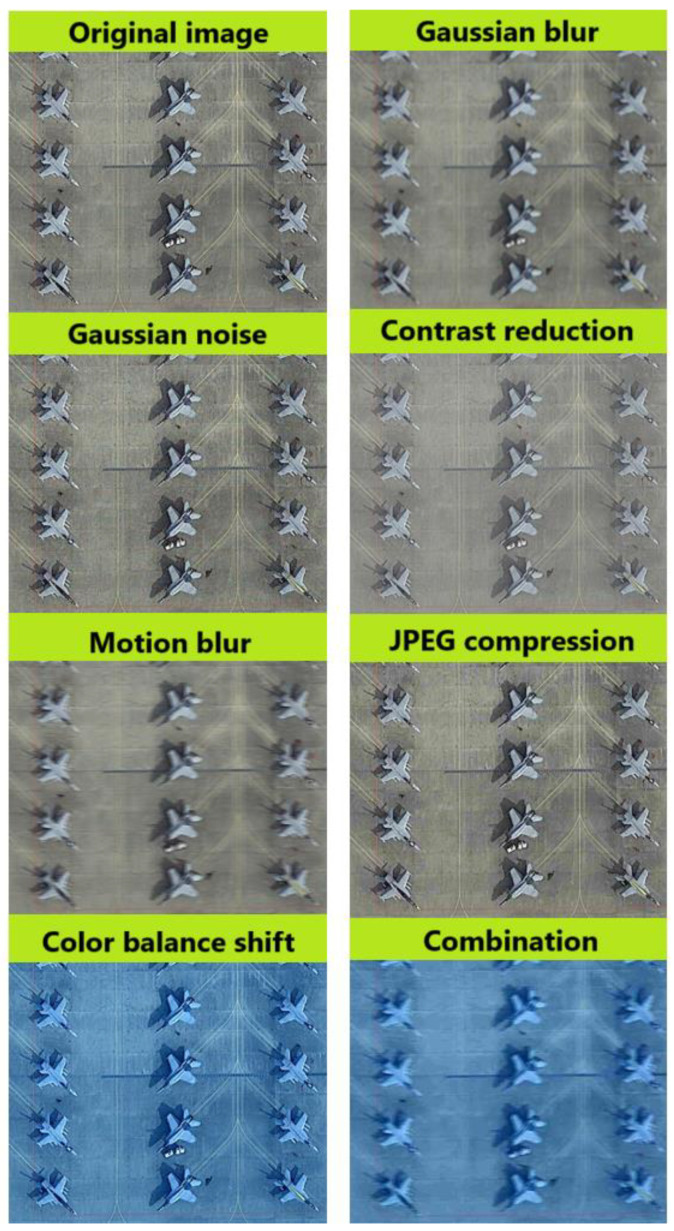
Interference simulating real operational conditions in image reconnaissance systems.

**Figure 5 sensors-24-07977-f005:**
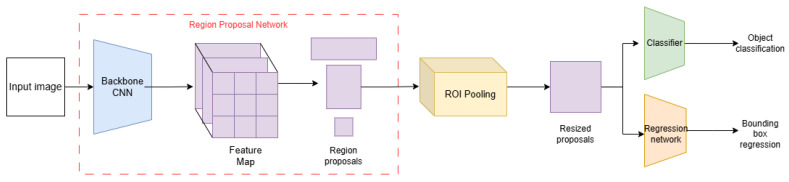
Diagram of the Faster R-CNN architecture [88,89].

**Figure 6 sensors-24-07977-f006:**
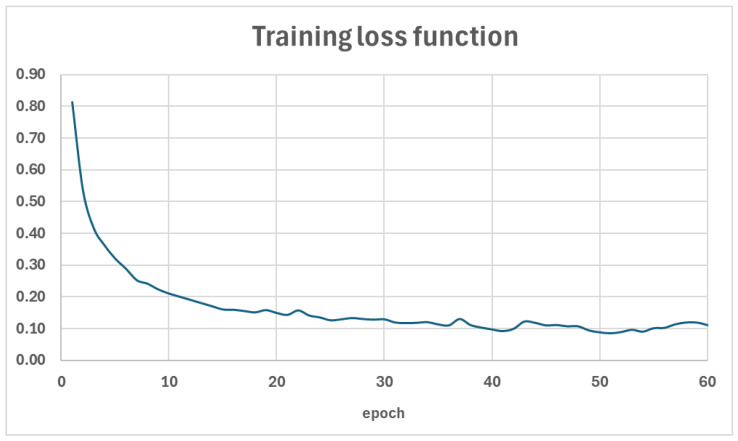
The value of the loss function in the training phase.

**Figure 7 sensors-24-07977-f007:**
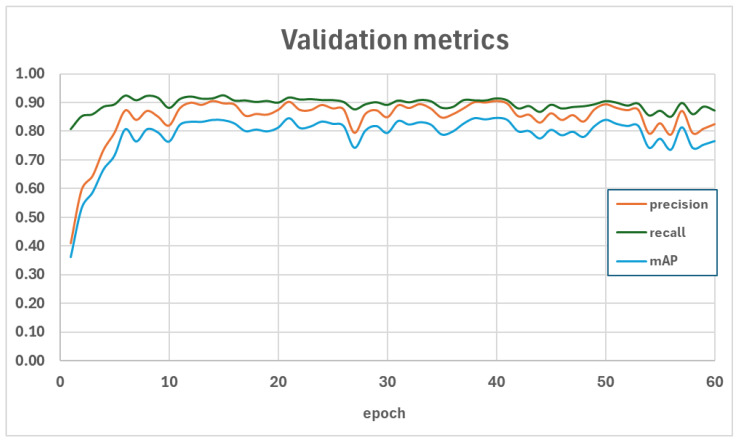
The values of the detection evaluation metrics in the validation phase.

**Figure 8 sensors-24-07977-f008:**
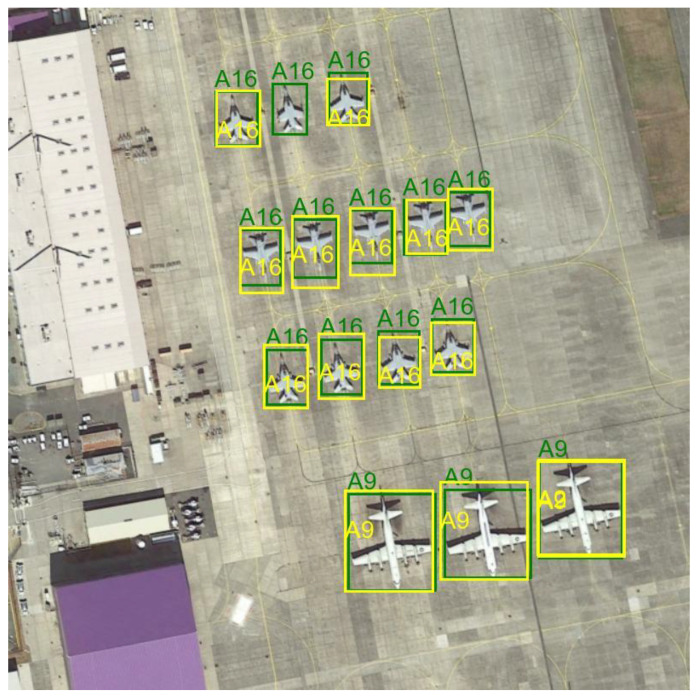
Detection model result—example 1 Green color indicates ground truth, while yellow color indicates the result of the model.

**Figure 9 sensors-24-07977-f009:**
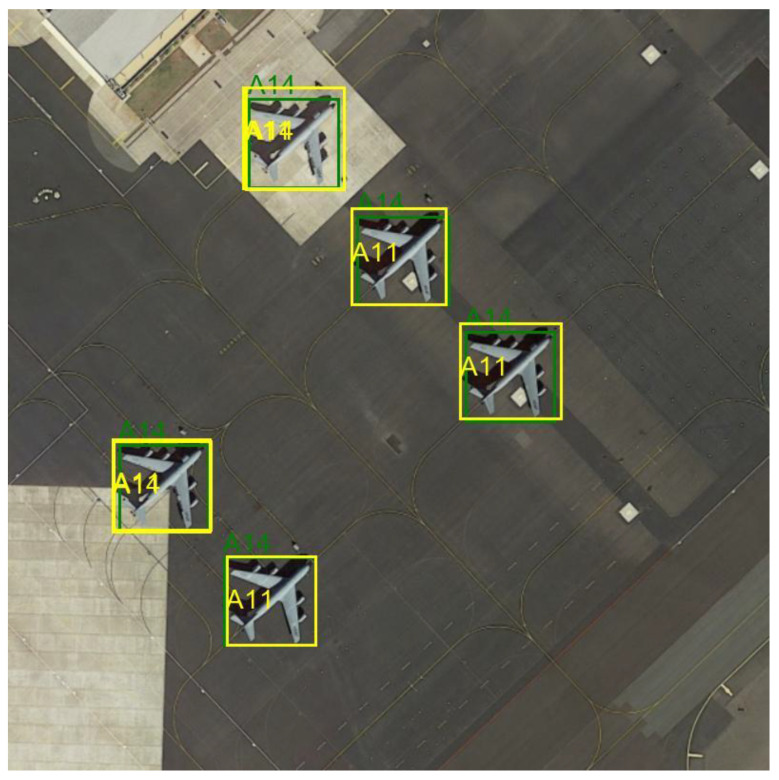
Detection model result—example 2. Green color indicates ground truth, while yellow color indicates the result of the model.

**Figure 10 sensors-24-07977-f010:**
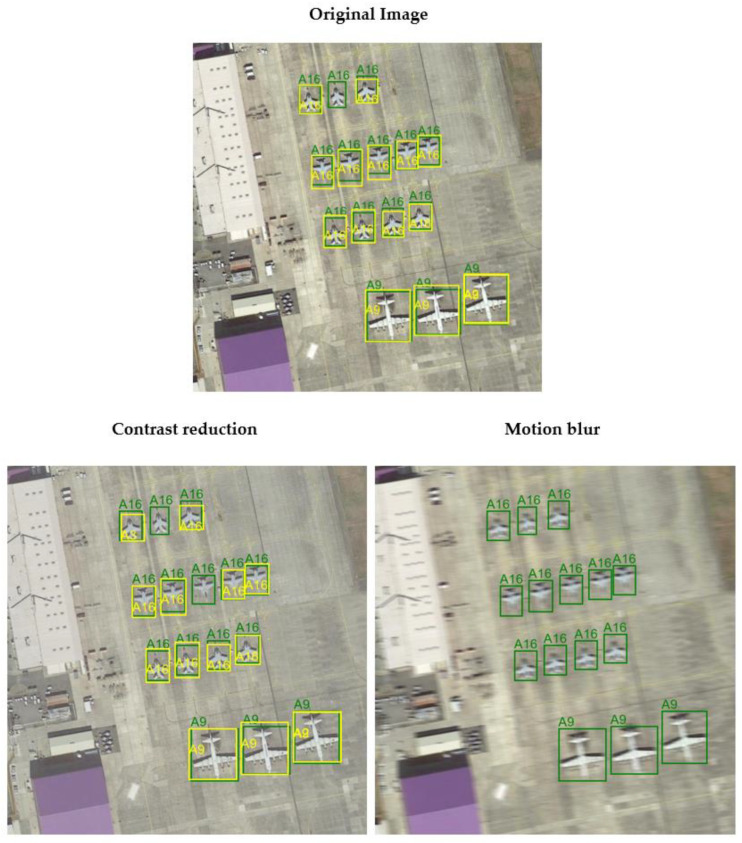
The influence of distortions on the performance of the detection model. Green color indicates ground truth, while yellow color indicates the result of the model.

**Figure 11 sensors-24-07977-f011:**
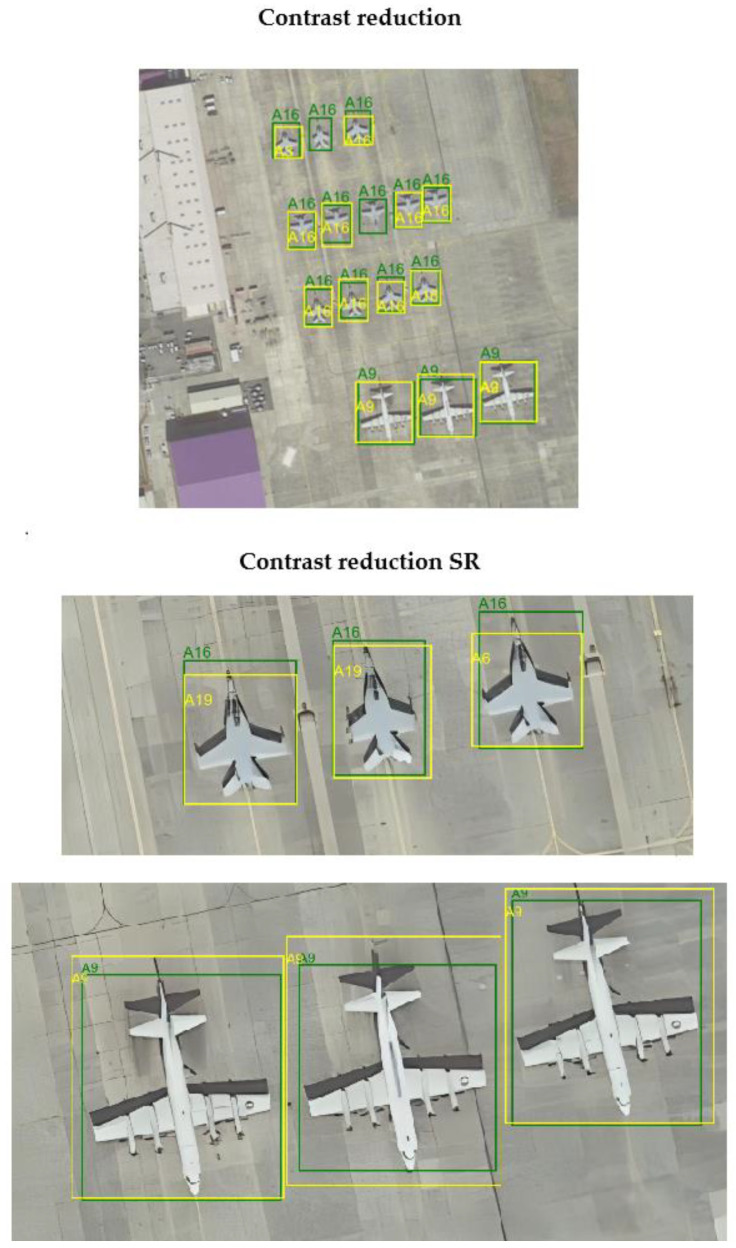
The influence of the super-resolution algorithm on the detection model performance—contrast reduction. Green color indicates ground truth, while yellow color indicates the result of the model.

**Figure 12 sensors-24-07977-f012:**
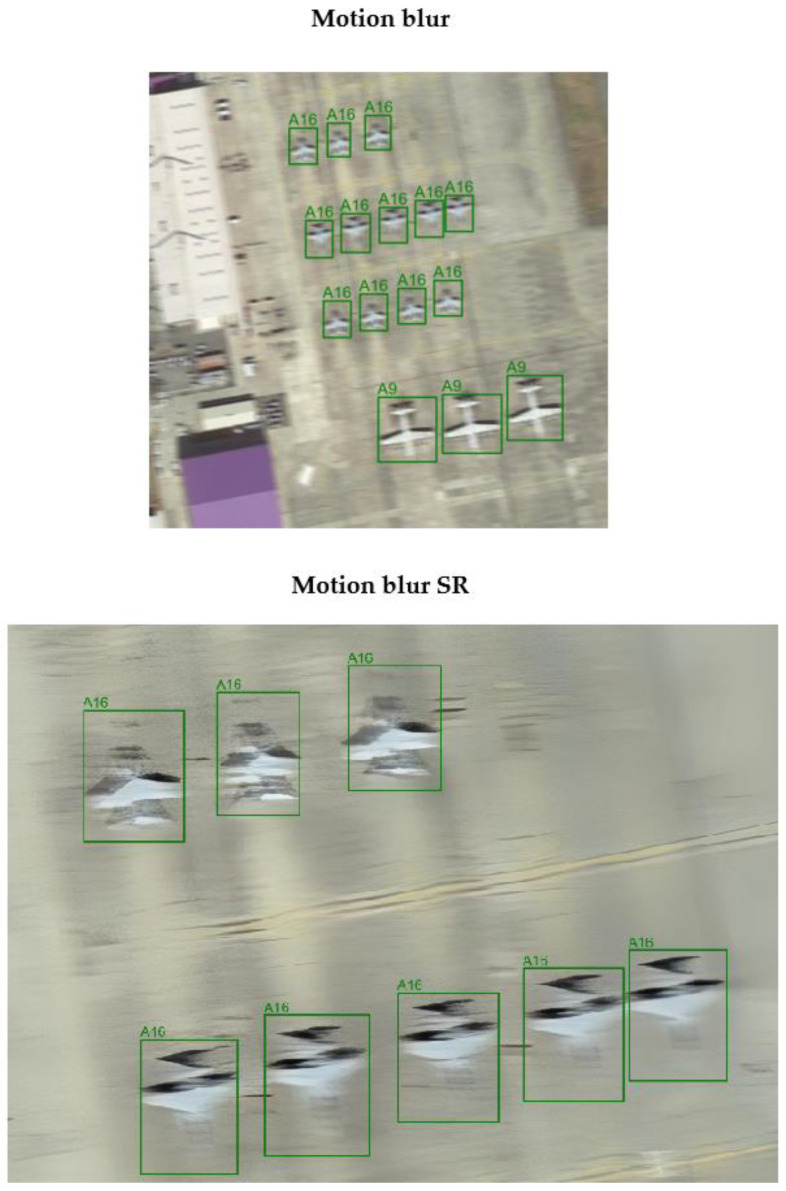
The influence of the super-resolution algorithm on the detection model performance—motion blur. Green color indicates ground truth, while yellow color indicates the result of the model.

**Figure 13 sensors-24-07977-f013:**
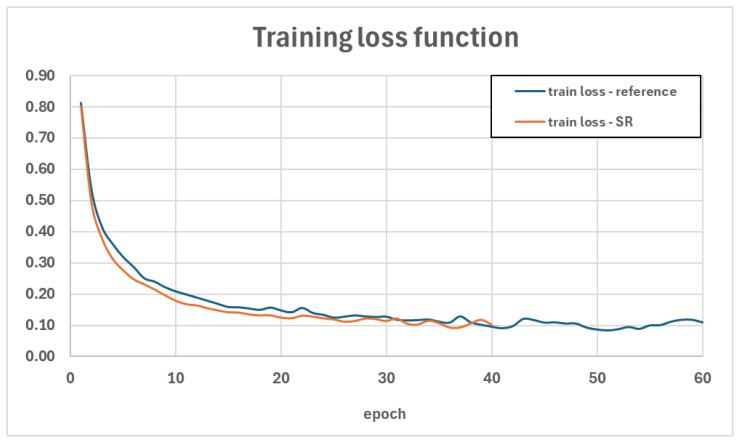
Comparison of loss function values in the training phase.

**Table 1 sensors-24-07977-t001:** Values of evaluation metrics for the test set.

Test Recall	Test Precision	Test mAP
0.847	0.630	0.596

**Table 2 sensors-24-07977-t002:** Summary of Average Precision and Average Recall for the test set.

	IoU	Objects Size	Max Detection Number	Value
Average Precision	0.50:0.95	all	100	0.596
Average Precision	0.50	all	100	0.630
Average Precision	0.75	all	100	0.630
Average Precision	0.50:0.95	small	100	−1.000
Average Precision	0.50:0.95	medium	100	0.755
Average Precision	0.50:0.95	large	100	0.568
Average Recall	0.50:0.95	all	1	0.341
Average Recall	0.50:0.95	all	10	0.820
Average Recall	0.50:0.95	all	100	0.847
Average Recall	0.50:0.95	small	100	−1.000
Average Recall	0.50:0.95	medium	100	0.778
Average Recall	0.50:0.95	large	100	0.904

**Table 3 sensors-24-07977-t003:** Values of evaluation metrics for distorted sets.

	Test Recall	Test Precision	Test mAP
Reference results	0.847	0.630	0.596
Gaussian blur	0.794	0.546	0.509
Gaussian noise	0.798	0.532	0.496
Contrast reduction	0.830	0.616	0.583
Motion blur	0.494	0.256	0.236
JPEG compression	0.812	0.555	0.519
Color balance shift	0.778	0.525	0.489
Combination	0.285	0.131	0.118

**Table 4 sensors-24-07977-t004:** Summary of Average Precision and Average Recall for Contrast Reduction Distortion.

	IoU	Objects Size	Max Detection Number	Value
Average Precision	0.50:0.95	all	100	0.583
Average Precision	0.50	all	100	0.616
Average Precision	0.75	all	100	0.616
Average Precision	0.50:0.95	small	100	−1.000
Average Precision	0.50:0.95	medium	100	0.746
Average Precision	0.50:0.95	large	100	0.566
Average Recall	0.50:0.95	all	1	0.333
Average Recall	0.50:0.95	all	10	0.804
Average Recall	0.50:0.95	all	100	0.830
Average Recall	0.50:0.95	small	100	−1.000
Average Recall	0.50:0.95	medium	100	0.749
Average Recall	0.50:0.95	large	100	0.912

**Table 5 sensors-24-07977-t005:** Summary of Average Precision and Average Recall for Motion Blur Distortion.

	IoU	Objects Size	Max Detection Number	Value
Average Precision	0.50:0.95	all	100	0.236
Average Precision	0.50	all	100	0.256
Average Precision	0.75	all	100	0.255
Average Precision	0.50:0.95	small	100	−1.000
Average Precision	0.50:0.95	medium	100	0.413
Average Precision	0.50:0.95	large	100	0.339
Average Recall	0.50:0.95	all	1	0.208
Average Recall	0.50:0.95	all	10	0.492
Average Recall	0.50:0.95	all	100	0.494
Average Recall	0.50:0.95	small	100	−1.000
Average Recall	0.50:0.95	medium	100	0.411
Average Recall	0.50:0.95	large	100	0.671

**Table 6 sensors-24-07977-t006:** Values of PSNR metrics for image pairs (averaged values for the entire subset of distortion data).

	Average PSNR [dB] (HR vs. LR)	Average PSNR [dB] (HR vs. SR)	Average PSNR [dB] (LR vs. SR)
Gaussian blur	34.37	32.78	35.19
Gaussian noise	28.93	32.96	28.84
Contrast reduction	30.42	29.63	34.68
Motion blur	33.90	32.28	35.04
JPEG compression	33.50	32.34	32.74
Color balance shift	29.19	28.57	33.91
Combination	28.42	28.48	34.27

**Table 7 sensors-24-07977-t007:** Values of evaluation metrics for restored sets.

	Test Recall	Test Precision	Test mAP
Reference results	0.847	0.630	0.596
Gaussian blur	0.794	0.546	0.509
Gaussian blur SR	0.825	0.560	0.521
Gaussian noise	0.798	0.532	0.496
Gaussian noise SR	0.831	0.590	0.553
Contrast reduction	0.830	0.616	0.583
Contrast reduction SR	0.854	0.591	0.555
Motion blur	0.494	0.256	0.236
Motion blur SR	0.416	0.193	0.172
JPEG compression	0.812	0.555	0.519
JPEG compression SR	0.838	0.575	0.534
Color balance shift	0.778	0.525	0.489
Color balance shift SR	0.801	0.530	0.496
Combination	0.285	0.131	0.118
Combination SR	0.256	0.115	0.101

**Table 8 sensors-24-07977-t008:** Values of evaluation metrics for the test set for both models.

	Test Recall	Test Precision	Test mAP
Original model	0.847	0.630	0.596
SR model	0.840	0.744	0.701

**Table 9 sensors-24-07977-t009:** Summary of Average Precision and Average Recall for the test set for model fine-tuned with restored data.

	IoU	Objects Size	Max Detection Number	Value
Average Precision	0.50:0.95	all	100	0.701
Average Precision	0.50	all	100	0.744
Average Precision	0.75	all	100	0.743
Average Precision	0.50:0.95	small	100	−1.000
Average Precision	0.50:0.95	medium	100	−1.000
Average Precision	0.50:0.95	large	100	0.701
Average Recall	0.50:0.95	all	1	0.333
Average Recall	0.50:0.95	all	10	0.820
Average Recall	0.50:0.95	all	100	0.840
Average Recall	0.50:0.95	small	100	−1.000
Average Recall	0.50:0.95	medium	100	−1.000
Average Recall	0.50:0.95	large	100	0.840

**Table 10 sensors-24-07977-t010:** Values of evaluation metrics for distorted and restored sets.

	Test Recall	Test Precision	Test mAP
SR model	0.840	0.744	0.701
Gaussian blur	0.728	0.509	0.469
Gaussian blur SR	0.811	0.641	0.595
Gaussian noise	0.755	0.574	0.532
Gaussian noise SR	0.820	0.694	0.656
Contrast reduction	0.807	0.662	0.625
Contrast reduction SR	0.816	0.688	0.649
Motion blur	0.403	0.224	0.207
Motion blur SR	0.269	0.127	0.114
JPEG compression	0.777	0.602	0.560
JPEG compression SR	0.829	0.679	0.637
Color balance shift	0.809	0.650	0.612
Color balance shift SR	0.821	0.697	0.656
Combination	0.365	0.180	0.162
Combination SR	0.327	0.150	0.134

## Data Availability

Data are available in a publicly accessible repository: Military Aircraft Recognition.

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
