# Peer review of "Optimization of Imaging Reconnaissance Systems Using Super-Resolution: Efficiency Analysis in Interference Conditions"

_sensors, 2024, doi:10.3390/s24247977_

Round 1

Reviewer 1 Report

Comments and Suggestions for Authors

This manuscript mainly discusses the impact of super-resolution technology, especially Real -ESRGAN, on detection performance under interference conditions. However, the authors have directly provided detection performance indicators. It is recommended to first evaluate the super-resolution performance indicators, that is, to first determine whether the super-resolution technology is effective, and then evaluate the impact on detection performance.

From the perspective of image features, Gaussian blur and motion blur are similar. It is suggested to further analyze and discuss why super-resolution is effective for Gaussian blur but not for motion blur. Additionally, the size of the blur kernel and whether it affects the results should be further discussed.

Besides, the authors argue that “In such cases, combining SR methods with traditional image enhancement techniques, such as filtering, or additional fine-tuning of the SR model specifically for certain distortionssuch as motion blurmay prove effective.” It is suggested to provide further verification.

The title of 5.2 “a augmentacja danych” should be in English.

Author Response

A detailed point-by-point response to all comments presented can be found in the attachment.

Reviewer 2 Report

Comments and Suggestions for Authors

1. The article addresses a crucial problem in the field of image reconnaissance systems, where the ability to detect and identify objects accurately is vital. 

However, there are few concerns.

2. While the study focuses on the Real-ESRGAN model, there is no discussion of other potential SR models or methods. 

3. The study relies on the Faster R-CNN detection model, which is not representative of all detection models in the field. Including a comparison with other models makes the results more generalizable to a wider range of applications.

4. The study focuses primarily on the theoretical and controlled application of SR to detection models, without incorporating real-world operational data or scenarios. This limits the practical applicability of the findings to real-world reconnaissance systems.

5. The study mentions distortions like motion blur, noise, and compression artifacts but does not explore how SR affects detection performance in other types of interference, such as lighting variations, occlusion, or environmental distortions. A more comprehensive analysis of various real-world interference types is valuable.

6. There is little analysis on the robustness of the SR-enhanced detection model across different levels of interference. For instance, the paper doesn’t mention if the improvements hold across a wide range of noise levels or motion blur intensity.

7. The results primarily focus on precision and mAP, but other evaluation metrics like recall, F1-score, or real-time detection accuracy are not considered. These additional metrics are important for understanding the overall performance of detection systems.

8. The study does not discuss the possibility of overfitting in the Faster R-CNN model after enhancing the data with SR. The enhanced images lead to overfitting in the detection model, reducing generalization to unseen or real-world data.

9. The research focuses on training and testing with a specific dataset. If the dataset does not sufficiently represent real-world variation (e.g., varied environments, diverse object types, etc.), the results do not fully translate to broader, real-world scenarios.

10. The paper does not address the limitations or potential drawbacks of using SR models in image reconnaissance systems, such as artifacts introduced by the SR process itself, or the difficulty of fine-tuning SR models for specific detection tasks.

11. The study does not compare the SR approach with other traditional image enhancement techniques, such as filtering or denoising, which are commonly used in reconnaissance systems. A comparison provides a clearer understanding of SR's relative advantages and disadvantages.

12. While SR improves detection accuracy, the paper does not consider the real-time processing constraints of SR models in operational settings. Image reconnaissance systems often require rapid processing, and SR techniques add latency or require significant computational power.

13. While the study mentions areas for future development, it does not clearly specify concrete research directions or practical steps for overcoming the identified challenges in SR-based image reconnaissance.

Author Response

(The authors gave the same response as above.)

Round 2

Reviewer 1 Report

Comments and Suggestions for Authors

The authors claim that despite using the same kernel size, the degradation effect of motion blur is much greater than that of Gaussian blur. It is recommended to provide detailed explanations or supplement relevant references.

Author Response

Dear Reviewer,

Thank you for pointing out the need for a more detailed explanation. In response to this comment, we have expanded the description of the distortions in Section 3.1 to better explain why motion blur causes more pronounced degradation than Gaussian blur, despite using the same kernel size.

We have added information to the manuscript that Gaussian blur is isotropic and uses a symmetric Gaussian distribution, leading to uniform degradation in all directions. This preserves some structural details of the image. Motion blur is directional, applying linear blurring along one axis, resulting in a significant loss of detail in that direction.

This distinction has now been clarified in the manuscript, and we hope it addresses your concerns.

We appreciate this suggestion, as it has allowed us to improve the clarity of our analysis.

Reviewer 2 Report

Comments and Suggestions for Authors

The revised manuscript addressed my concerns.

Author Response

Dear Reviewer,

Thank you very much for your positive feedback and for taking the time to review our revised manuscript. We greatly appreciate your insightful comments and suggestions, which helped us improve the quality and clarity of the paper. Your observations also drew our attention to certain aspects that required further refinement.

We are particularly grateful for the numerous suggestions you provided, which will undoubtedly guide us as we continue to explore this research topic in future studies.

Thank you again for your valuable contributions to the development of our work.